# XYZ-IBD: Benchmarking Robust 6D Object Pose Estimation under Real-World Industrial Complexity

## Abstract

We introduce XYZ-IBD, a bin-picking benchmark for 6D pose estimation that captures real-world industrial complexity, including challenging object geometries, reflective materials, severe occlusions, and dense clutter. The dataset reflects authentic robotic manipulation scenarios with millimeter-accurate annotations. Unlike existing datasets that primarily focus on household objects, which approach saturation, XYZ-IBD represents the unsolved vision problems in the real-world application. The dataset features metallic and mostly symmetrical objects of varying shapes and sizes. These objects are heavily occluded and randomly arranged in bins with high density, replicating the challenges of industrial bin-picking. XYZ-IBD was collected using two high-precision industrial cameras and one commercially available camera, providing RGB, grayscale, and depth images. It contains 75 multi-view real-world scenes with around 273k annotated object instances, along with a large-scale synthetic dataset rendered under simulated bin-picking conditions. We employ a meticulous annotation pipeline that includes anti-reflection spray, multi-view depth fusion, and semi-automatic annotation, achieving millimeter-level pose labeling accuracy required for industrial manipulation. Quantification in simulated environments confirms the reliability of the ground-truth annotations. We benchmark state-of-the-art methods on 2D detection and 6D pose estimation tasks on our dataset, revealing significant performance degradation in our setups compared to current academic household benchmarks. By capturing the complexity of real-world bin-picking scenarios, XYZ-IBD introduces more realistic and challenging vision problems for future research.

## 1 Introduction

The ability to detect, segment, and estimate the 6D pose of objects is critical for robotics applications, particularly in industrial bin-picking scenarios. These tasks demand not only high accuracy but also efficiency to enable real-time operation. While recent advancements in computer vision have significantly improved performance on benchmark datasets Sundermeyer et al. (2023), there remains a substantial gap between academic research and real-world applications Van Nguyen et al. (2025); bop. This discrepancy is especially pronounced in industrial settings, where challenges such as clutter, occlusion, and reflective, texture-less objects must be addressed.

Current popular benchmarks for pose estimation, such as those designed for household objects Xiang et al. (2018); Kaskman et al. (2019); Hodan et al. (2018), often exhibit favorable properties including rich textures, semantic cues, low occlusion, and minimal clutter. Some datasets Doumanoglou et al. (2016); Hodaň et al. (2017); Brachmann et al. (2014) have extended the challenge by introducing texture-less objects Hodaň et al. (2017), cluttered scenes Brachmann et al. (2014), or robotic bin-picking setups Doumanoglou et al. (2016). While these household benchmarks have driven significant progress in pose estimation pipelines, state-of-the-art methods Wen et al. (2024); Lin et al. (2024); Wang et al. (2021) still struggle with industrial objects that are highly reflective, symmetric, or lack distinctive visual features Drost et al. (2017). Unlike household objects, industrial items often lack contextual semantics and present ambiguous appearances, making them particularly difficult for feature extraction and accurate pose estimation, thereby the methods falling short of the precision requirements in real-world industrial manipulation. Although several datasets have begun to address

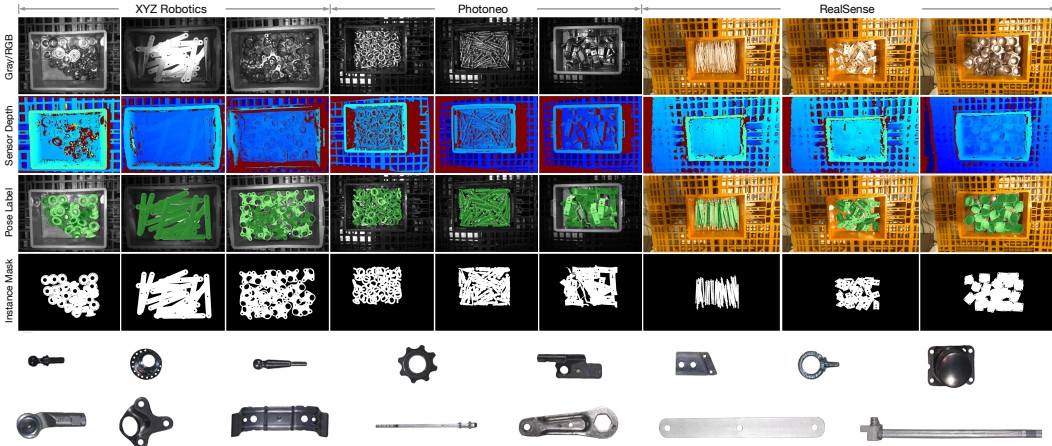

Figure 1: Example data from our industrial bin-picking dataset that shows challenging scenes captured by three cameras with different modalities, along with our 6D pose annotations. The industrial parts adopted in this benchmark present variant geometry and size.

these challenges by including texture-less, reflective, and symmetric industrial objects Hodaň et al. (2017); De Roovere et al. (2022); Liu et al. (2021); Kalra et al. (2024), they still lack configurations that fully replicate the complexity of industrial bin-picking scenarios. These include randomly stacked objects in containers, harsh and variable lighting conditions, diverse object geometries, and multiple repeated instances with severe occlusion.

To address this need, we introduce XYZ-IBD, a novel RGB-D dataset specifically designed for industrial bin-picking applications. Unlike existing datasets, XYZ-IBD captures the complexity of real-world industrial environments, including challenging object geometries, severe scene clutter, and strong spectral reflections. The dataset features 15 texture-less, metallic, and mostly symmetric objects commonly used in industrial settings. As shown in Figure 1, these objects vary in shape and size, and are densely packed with multiple instances in cluttered bins, creating significant occlusion. To ensure diverse and practical data modalities, we capture multi-view RGB and depth images using two high-precision industrial-grade cameras, an XYZ Robotic DLP structured light camera, and a Photoneo PhoXi laser scanner alongside a commercially available Intel RealSense D415 stereoscopic camera. The dataset consists of 75 real-world scenes (5 configurations per object), comprising over 22k labeled multi-view RGB-D frames and approximately 273k 6D pose annotations. Additionally, we provide a large-scale synthetic training set containing up to 45k rendered views generated using BlenderProc Denninger et al. (2020), simulating realistic bin-picking conditions through physics-based object interactions.

Given the millimeter-level precision required for industrial bin-picking, ensuring the accuracy of these annotations is essential. To provide accurate 6D pose annotations, we employ a multi-step, semi-automatic annotation pipeline. First, we sample and calibrate multiple viewpoints within a specified working distance using four calibration spheres. To enable precise depth map acquisition, we apply an anti-reflection spray to the objects, following practices from prior work Yang et al. (2021); Jung et al. (2023; 2024). We then fuse the depth data from multiple views using the high-quality ground truth depth. A self-developed annotation tool is used to label each object instance in the fused point cloud. Finally, the annotated object poses from the reference frame are projected to all remaining frames, followed by a manual double-check pass.

To quantify the accuracy of our pose annotations, we simulate real-world setups within a controlled simulation environment. We replicate the exact camera intrinsics and extrinsics used in our real-world experiments and randomly arrange objects in a virtual container. To closely mirror real-world conditions, we introduce camera measurement noise into the simulated images. The same annotation pipeline used for the real dataset is then applied to the simulated scenes. By comparing the resulting annotations to the ground truth poses available in the simulation, we compute the annotation error. This evaluation validates that our annotations are precise enough to serve as reliable ground truth for benchmarking 6D pose estimation methods. The overall data collection and annotation quantification is illustrated in Figure 2.

**Real-world Data Acquisition**

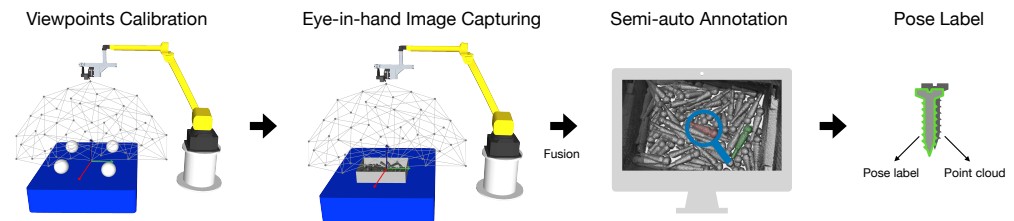

**Annotation Quantification**

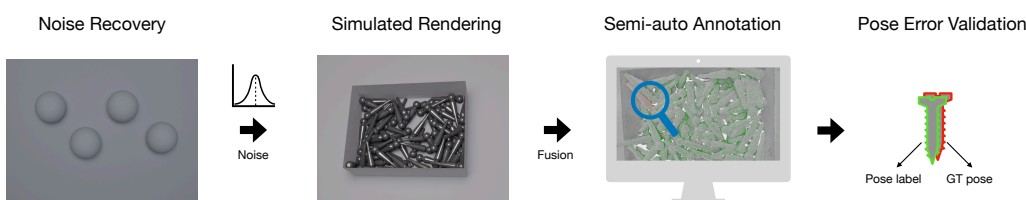

Figure 2: The real-world industrial data collection pipeline and the annotation error quantification pipeline in the simulated environment for XYZ-IBD dataset.

We benchmark XYZ-IBD on object 2D detection and object 6D pose estimation tasks with state-of-the-art methods. While these methods perform exceptionally well on existing datasets, our experiments reveal a stark performance drop in the challenging conditions posed by XYZ-IBD. This highlights the current gap between recent academic benchmarks and real industrial conditions. By addressing the challenging scenarios, XYZ-IBD provides a much-needed benchmark for advancing detection and pose estimation methods in realistic industrial settings. We firmly believe XYZ-IBD bridges the gap between current academic benchmarks and practical vision problems, fostering the development of more robust and efficient solutions for industrial robotics, ultimately improving automation and machine support.

In summary, our key contributions are :

- We build a challenging real-world industrial bin-picking dataset that simultaneously captures the complexity of object geometry, material, occlusion, and clutter, introducing academic challenges for object detection and pose estimation tasks.

- Our dataset provides high-quality annotation for the industrial-grade demanded millimeter-level precision that provides accurate labels for reliable evaluation.

- We benchmark the dataset with 2D detection and 6D detection tasks with the most recent baselines, for both instance-specific and generalizable frameworks.

## 2 RELATED WORK

### 2.1 HOUSEHOLD DATASETS

A large number of object pose and scene depth datasets have been developed to address everyday scenarios involving household objects. Datasets such as LineMOD Hinterstoisser et al. (2012), LineMOD-Occlusion Brachmann et al. (2014), YCB-V Xiang et al. (2018), Home-brewedDB Kaskman et al. (2019), and TUD-L Hodan et al. (2018) are widely used in benchmarks for model-based object pose estimation Van Nguyen et al. (2025); Hodan et al. (2018), and have driven progress on key challenges such as handling texture-less objects Hinterstoisser et al. (2012), occluded targets Brachmann et al. (2014), and typical household environments Xiang et al. (2018); Kaskman et al. (2019). The HOPE dataset Tyree et al. (2022) extends this focus to robotic manipulation scenarios with varied lighting and occlusion conditions. IC-BIN Doumanoglou et al. (2016) introduces an early bin-picking setup with randomly placed objects, but it includes only two textured objects and

Table 1: Comparison of datasets for object pose estimation from different dimensions.

| Dataset | Modalities | Number of Objects | Object Diversity | Object Diameter (mm) | Frames | Instances | Instances per Scene | Accurate Depth GT | Occlusion | Reflection | Labeling Error (mm) |
|---|---|---|---|---|---|---|---|---|---|---|---|
| DIMO De Roovere et al. (2022) | RGB-D | 6 | + | 75~302 | 31.2k | 100k | <10 | ✗ | + | ✗ | 2.7 |
| T-LESS Hodaň et al. (2017) | RGB-D | 30 | +++ | 63~152 | 147k | 100k | <10 | ✗ | ++ | ✗ | 11.3 |
| ITODD Drost et al. (2017) | RGB-D | 28 | +++ | 24~270 | 800 | 5k | <10 | ✗ | ++ | ✗ | 1.8 |
| ROBI Yang et al. (2021) | RGB-D | 7 | + | 24~76 | 8k | 600k | >10 | ✓ | +++ | ✓ | 1.8 |
| StereoOBJ-1M Liu et al. (2021) | RGB | 18 | ++ | - | 396k | 1.5M | <10 | ✗ | + | ✗ | 2.3 |
| IPD Kalra et al. (2024) | RGB-D+Polar | 20 | +++ | 80~240 | 30k | 100k | <10 | ✗ | + | ✓ | N/A |
| XYZ-IBD (Ours) | RGB-D | 15 | +++ | **54~300** | 22.5k | 273k | **22** | ✓ | +++ | ✓ | **0.99** |

suffers from low annotation quality. StereoOBJ-1M Liu et al. (2021) improves annotation precision through structure-from-motion (SfM) with checkerboards and offers a large number of RGB images, yet it lacks object diversity and does not include depth data. NOCS Wang et al. (2019) presents the first category-level 6D pose dataset, covering six household object categories. More recent datasets such as PhoCal Wang et al. (2022), HouseCat6D Jung et al. (2022), Booster Ramirez et al. (2023), and SCRREAM Jung et al. (2024) focus on more complex scenes involving transparent or highly reflective objects and utilize a range of sensor modalities, including RGB, depth, and polarization images. While those datasets provide high-quality depth and pose annotations, they lack typical scene properties found in industrial environments. Therefore, existing datasets featuring household objects do not fully capture the challenges inherent in industrial applications, which involve both object-level and scene-level complexity.

## 2.2 INDUSTRIAL DATASETS

In industrial applications, the working environment is quite different from the household scenario. Firstly, unlike household objects, industrial parts are usually texture-less and often symmetric and highly reflective Drost et al. (2017); Kalra et al. (2024); Yang et al. (2021). Consequently, networks trained on household objects hardly generalize to industrial datasets. Secondly, the required pose accuracy in industrial robotics is usually higher than in household robotics or AR/VR applications. The robotic arm is expected to not only pick up singulated objects, but typically needs to pick objects from a filled container and place them at a target pose or assemble them. Even though bin-picking is a typical setup for industrial applications, only a few publicly available datasets target this scenario which severely hampers the usability of pose estimation pipelines in industrial practice. T-LESS dataset Hodaň et al. (2017) features texture-less industrial objects with symmetries but does not present challenging lighting conditions, and the annotation quality is not mm-accurate, a requirement in many industrial applications. Only a few scenes present the complexity of real bin-picking configurations where similar objects occlude each other. ITODD Drost et al. (2017) collects industrial parts with challenging geometry and lighting conditions but does not feature bins filled with objects. The consistently low pose estimation scores on ITODD Drost et al. (2017) in the BOP challenge Van Nguyen et al. (2025) also demonstrate the need for industrial bin-picking datasets. Other datasets such as DIMO De Roovere et al. (2022) and ROBI Yang et al. (2021) focus on metallic objects for bin-picking setups, but they focus on a limited number of objects whose size and shape are not representative of the diversity in real applications. The recent dataset IPD Kalra et al. (2024) leverages multiple sensors to collect data from industrial objects but presents little clutter, stacking and occlusions which simplifies the setup compared to real industrial scenarios. Table 1 compares the characteristics of current industrial datasets.

## 3 THE XYZ-IBD DATASET

XYZ-IBD establishes a benchmark for industrial bin-picking by capturing data under authentic factory conditions. It advances prior work through four perspectives: **(1) Industrial-Grade Setup**: Data is acquired using industry-standard robotic arms (FANUC M10iD/8L) and multi-modal sensors (RGB/depth/grayscale) mounted at industrial working distances, replicating real application conditions. **(2) Challenging Objects**: fifteen reflective, textureless, and mostly symmetric industrial parts that present rich geometrical shapes and sizes(54–300 mm scale), introducing academic challenges for pose estimation. **(3) Multi-instance Dense Clutter**: Objects are randomly and densely arranged in a container with multiple repeat instances, creating more ambiguity for instance detection and

alignment. **(4) Precise Annotation**: Our annotation pipeline achieves <1 mm positional and <1° angular annotation accuracy, validated with simulated environment.

This benchmark consists of **75 real-world scenes** (five configurations per object), encompassing approximately **273k annotated instances** with **22 instances per scene on average**, and **up to 60 instances in some scenes**. In addition, it includes **45k synthetic training set** generated with BlenderProc Denninger et al. (2020) through physics-based object interactions, simulating a realistic bin-picking setup.

### 3.1 OBJECTS AND HARDWARES

**Objects.** Our dataset comprises fifteen representative industrial parts with diameters ranging from 54 mm to 300 mm, including components like sheet metal parts, bolts, pins, covers, and many other kinds of machined metal objects. As shown in Figure 1, these objects exhibit challenging visual properties such as high reflectivity and symmetry that are common in manufacturing environments yet problematic for vision algorithms. The original CAD models provided by industrial partners ensure micron-level geometric accuracy for both real-world captures and synthetic renderings. All real-world data is collected in bins with sensor-to-object distances carefully calibrated between 600-1000mm. We put multiple instances of each object into the bin, mostly with severe occlusion and clutter.

**Sensor Setup.** For precise and repeatable data acquisition, we employ an industrial-grade FANUC M10iD/8L robotic arm with ±0.06mm repeatability to position our multi-sensor array. Three complementary vision systems are rigidly co-mounted on the end-effector (see Figure 6): the *Intel RealSense D415 stereoscopic camera* provides aligned RGB (1920×1080) and depth streams at 30 FPS, offering baseline color-depth registration for general scene understanding; the *XYZ Robotic AL-M DLP structured-light* camera delivers high-precision grayscale (1440×1080) and depth maps (0.08mm resolution) through projected pattern deformation analysis, particularly effective for matte surfaces; the *Photoneo PhoXi M 3D scanner* utilizes laser triangulation to generate high-accuracy depth data (up to 2064×1544 resolution) with 0.1 mm voxel precision, complemented by synchronized grayscale imagery. All sensors are positioned at optimized working distances of 600-1000 mm based on object size and bin geometry, maintaining consistent fields-of-view across the industrial container. The fixed relative positions between cameras enable direct cross-modality calibration, while the robotic arm's precise positioning ensures reproducible viewpoint acquisition throughout the data collection process.

### 3.2 DATA ACQUISITION PIPELINE.

As shown in Figure 2, our data acquisition pipeline integrates three sequential stages: viewpoint sampling and calibration, multi-pass scene capture for depth ground truth, and a hybrid annotation protocol combining manual and algorithmic refinement.

**Multi-view Sampling and Calibration.** Beginning with the bin's centroid as the origin, we define a spherical sampling surface spanning elevation angles of 45° to 90° to balance perspective diversity and robotic arm operability. Fifty viewpoints are randomly distributed across this surface to ensure comprehensive spatial coverage. Following the calibration framework of Yang et al. (2021), we place four precisely machined calibration spheres on the working plane. During an initial calibration pass, the robotic arm captures multi-modal images of these spheres across all viewpoints. The cameras are firstly undistorted and obtain the initial camera poses with hand-eye calibration Daniilidis (1999), then pose refinement via iterative closest point (ICP) alignment on the spheres' point clouds establishes relative transformations between viewpoints with around 0.248 mm average root mean square error (RMSE), resolving the 6 DoF relationships between 49 secondary viewpoints and a primary reference view. These transformations enable subsequent multi-view data fusion and label projection with sub-millimeter consistency. After calibration, the calibration spheres are removed, and the robotic arm systematically revisits each pre-calibrated viewpoint to capture cluttered industrial bin-picking scenes. At each viewpoint, three rigidly mounted cameras (Intel RealSense D415, XYZ Robotic DLP, Photoneo PhoXi) acquire synchronized RGB, grayscale, and depth data. To maximize scene diversity, we perform five complete capture cycles per object, randomly shuffling parts between cycles.

**Multi-Pass Scene Capture.** To address depth sensing challenges from reflective surfaces, we employ a dual-phase capture strategy. The first phase applies a temporary anti-reflective coating (Acksys SP-102) to suppress specularity, enabling high-fidelity ground truth depth acquisition. After

Raw Image | Anti-reflection Image | Raw Depth | Anti-reflection Detph

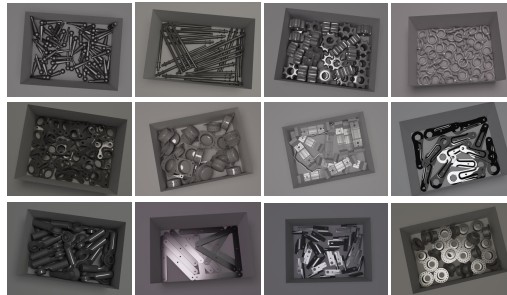

Figure 3: Comparison between raw depth and anti-reflection depth.

Figure 4: Example data of the synthetic training data with bin-picking simulation.

allowing 15 minutes for complete evaporation under controlled ambient conditions (25°C ±1°C), we execute an identical second capture pass to record the scene's native optical properties. Both phases maintain pixel-wise spatial correspondence through robotic arm pose repetition (±0.06 mm precision), providing high-quality depth to fuse the scene point cloud, thus resulting in more accurate pose annotation and also aligned datasets of enhanced and raw depth for algorithm benchmarking.

**6D Pose Annotation.** The annotation derives from a hierarchical process beginning with 3D fusion of spray-enhanced depth data into a unified scene point cloud. Annotators coarsely align CAD models to this reconstruction using our developed constrained GUI (±1 mm translational, ±1° rotational increments), followed by multi-scale ICP refinement. The ICP pipeline first aligns downsampled point clouds for global adjustment and then iteratively optimizes with full-resolution data to achieve sub-millimeter accuracy. Finalized poses are propagated to all 50 viewpoints using pre-calibrated transformations that we obtained from the first stage, ensuring label consistency across perspectives without manual per-view annotation. This protocol yields around 273k annotated object instances in total, with a minimum of 11 and a maximum of 60 instances per scene, resulting in an average of 22 instances per image. We choose the XYZ Robotic DLP camera as the primary camera and perform annotations on the data it collected. The annotation of the other two cameras are projected through the calibrated relative transformation between the cameras.

### 3.3 Annotation Error Quantification.

To quantify the cumulative annotation error, we replicated the data collection and annotation process in a simulated environment, recovering the sensor error, calibration error and the human annotation error, and compared the resulting annotations against ground truth poses. Our evaluation framework comprises three stages: data noise recovery, synthetic data collection, and 6D pose quantification.

**Data Noise Recovery.** We simulate the multi-view calibration procedure in a synthetic environment using identical calibration spheres and camera parameters as in the real-world setup. 50 calibration views are sampled, and varying levels of Gaussian noise are added to the rendered depth images. The corresponding RMSE of the point cloud is then computed using ICP, revealing the relationship between noise magnitude and calibration error (see Table 3). When Gaussian noise $\sigma = 0.26$ mm is applied, the computed RMSE reached 0.248 mm, matching the error observed during real-world calibration. This provided the chosen noise level to best represent the cumulative error introduced by the sensor, robotic system, and multi-view calibration process.

**Synthetic Data Collection.** To ensure realistic, cluttered arrangements, we generate synthetic counterparts using the same CAD models within a simulated bin-picking environment rendered with physically-based rendering in BlenderProc Denninger et al. (2020). Objects are randomly dropped into the bin via a free-fall simulation, and any that fall outside the bin are removed. Multi-view synthetic images are rendered using a complementary noise model derived in the last step. The same annotation pipeline used for real data, incorporating multi-view fusion, manual adjustments, and multi-scale ICP refinement, is also applied to the synthetic scenes. As ground truth poses are available in the simulation, this setup allows for direct comparison between annotated and true object poses.

To complement the real dataset, we additionally render a large-scale synthetic dataset as the training data. We programmatically vary rendering conditions, including lighting, material properties, object quantity, and pose configurations, closely replicating real-world setups to ensure cross-domain

Table 2: Specifications for data collection error sources.

| Source | Specify | Error (mm) |
|---|---|---|
| Viewpoints Calibration | Sensor Calibration | 0.10 |
| | Sensor Temporal noise | 0.10 |
| | Sensor Distortion | N/A |
| | Robot arm Repeatability | 0.06 |
| | Viewpoints Calibration RMSE (Total) | **0.245** |
| Depth fusion | TSDF | N/A |
| Manually annotate | Human, ICP | N/A |
| Overall | | **0.99** |

Table 3: Effect of Gaussian noise on average viewpoints calibration RMSE.

| Gaussian Noise $\sigma$ (mm) | RMSE (mm) |
|---|---|
| 0.0 | 0.199 |
| 0.1 | 0.209 |
| 0.2 | 0.233 |
| **0.26** | **0.248** |
| 0.3 | 0.259 |

consistency. For each of the 15 objects, we perform 120 free-fall simulations, with random variations in material and lighting, resulting in a total of approximately 45,000 frames in the synthetic training dataset. As shown in Figure 4, this parallel real-synthetic collection supports robust benchmarking while preserving strong visual alignment between domains.

**6D Pose Quantification.** To assess annotation quality, we systematically investigated several error sources: inherent sensor inaccuracies, robotic arm repeatability, viewpoint calibration discrepancies, and annotator subjectivity. Specifically, the sensor error encompasses both the camera calibration inaccuracies and measurement noise due to sensor characteristics and environmental conditions. This, together with the robot arm repeatability, is manifested in the overall multi-view pose calibration error. We compute pose errors through nearest-neighbor matching between annotated and GT poses using Hungarian assignment on 3D centroid distances. We analyze up to 60 samples per scene $\times$ 3 scenes per object, revealing a mean positional error of 0.999 mm ($\sigma = 0.12$ mm) and an angular error of $0.432°$ ($\sigma = 0.08°$). Per-object error averages across 15 industrial parts demonstrate sub-millimeter precision even for challenging geometries. This synthetic validation confirms that our real-world annotations achieve <1 mm positional and <1° angular accuracy relative to physical GT.

## 4 BENCHMARKS

Our dataset provides high-precision 6D object pose and depth annotations, enabling the establishment of a comprehensive benchmark for object detection and pose estimation. In order to align the evaluation protocol with the widely used Benchmark for 6D Object Pose Estimation (BOP) and challenges, we adopt their evaluation protocols to assess performance on our dataset. For the 2D detection and 6D pose estimation tasks, we evaluate representative methods under both seen and unseen object settings. In the seen object setup, models are trained on our synthetic dataset and evaluated on the real-world test split. In the unseen object setup, we directly use off-the-shelf generalizable methods, which have been pretrained on large-scale external datasets, to infer our real test scenes without extra finetuning. We benchmark several recent state-of-the-art methods across **five previous BOP-Core datasets** (LM-O Brachmann et al. (2014), T-LESS Hodaň et al. (2017), YCB-V Doumanoglou et al. (2016), IC-BIN Doumanoglou et al. (2016), TUD-L Hodan et al. (2018)) in compare with XYZ-IBD, under **four tasks** (model-based seen/unseen object 2D detection and model-based seen/unseen object 6D detection). *Detailed data splits and the implementation for these baseline methods are provided in the Appendix D.*

### 4.1 EVALUATION CRITERIA

**Object 2D Detection Metics.** For the object 2D detection task, we follow the model-based 2D detection task defined in BOP 2024-2025 Challenge Van Nguyen et al. (2025). The objective is to generate a set of non-overlapping 2D binary instance masks with associated confidence scores from an RGB-D input image that contains multiple object instances from a given dataset. To evaluate performance, we adopt the Average Precision (AP) metric, following the protocol used in the COCO 2020 challenges Lin et al. (2014). AP is calculated by averaging the precision scores at several Intersection-over-Union (IoU) thresholds, ranging from 0.5 to 0.95 in increments of 0.05. Each object's AP score reflects its detection quality across these thresholds. To obtain an overall dataset-level performance measure, the mean Average Precision (mAP) is computed by averaging the AP scores across all object categories. This evaluation strategy comprehensively captures both the accuracy of object localization and the effectiveness of category-level recognition, ensuring alignment with established benchmarking standards.

**Object 6D Detection Metics.** For the 6D pose estimation task, we adopt the model-based 6D object detection metric defined in BOP 2024-2025 Challenge Van Nguyen et al. (2025), evaluating detection accuracy using symmetry-aware Average Precision (AP) scores. For each predicted pose $\hat{P}$ and its corresponding ground truth pose $P_{GT}$, we compute two error metrics: *Maximum Symmetry-Aware Surface Distance (MSSD)* and *Maximum Symmetry-Aware Projection Distance(MSPD)*. MSSD measures the maximum 3D surface deviation under object symmetries, defined as $e_{\text{MSSD}} = \max_{x \in M} \min_{S \in S} |\hat{P}x - S(P_{GT}x)|$, where $M$ is the object mesh and $S$ is the set of predefined symmetry transformations. MSPD evaluates the maximum 2D projection deviation considering object symmetries, computed as $e_{\text{MSPD}} = \max_{u \in U} \min_{S \in S} |\Pi(\hat{P}x_u) - \Pi(S(P_{GT}x_u))|$, where $\Pi$ denotes the camera projection function and $U$ the set of visible mesh vertices. A pose estimate is deemed correct when the error $e$ falls below a threshold $\theta_e$. For each error type $e \in \text{MSSD}, \text{MSPD}$ and object $o \in O$, we compute the object-level AP score as $AP_{e,o} = \frac{1}{|\Theta_e|} \sum_{\theta \in \Theta_e} P_o(\theta)$, where $\Theta_e$ is the set of threshold values and $P_o(\theta)$ is the precision at threshold $\theta$. The final AP score aggregates over all objects and both error types as $\text{AP} = \frac{1}{2|O|} \sum_{o \in O} \sum_{e \in \text{MSSD,MSPD}} AP_{e,o}$.

Table 4: Performance comparison of 2D and 6D detection SOTA methods on previous core datasets in BOP and XYZ-IBD dataset. We report the AP scores on the main BOP datasets YCB-V, T-LESS, and LM-O, and the average AP on the 5 BOP core datasets (YCB-V, T-LESS, LM-O, IC-BIN and TUD-L) under the four tasks of seen object 2D detection, unseen object 2D detection, seen object 2D detection, and unseen object 2D detection.

| Method | Task | Unseen Object | YCB-V | T-LESS | LM-O | BOP-Core 5 | XYZ-IBD |
|---|---|---|---|---|---|---|---|
| YOLOX Ge et al. (2021) | | ✗ | 0.877 | 0.707 | 0.894 | 0.798 | 0.774 |
| CNOS(SAM) Nguyen et al. (2023) | 2D Detection | ✓ | 0.490 | 0.395 | 0.330 | 0.361 | 0.275 |
| SAM-6D(SAM) Lin et al. (2024) | | ✓ | 0.518 | 0.465 | 0.437 | 0.449 | 0.296 |
| GDRN Wang et al. (2021) | | ✗ | 0.906 | 0.852 | 0.775 | 0.827 | 0.266 |
| SurfEmb Haugaard & Buch (2022) | 6D Detection | ✗ | 0.799 | 0.828 | 0.760 | 0.758 | 0.247 |
| FoundationPose Wen et al. (2024) | | ✓ | 0.889 | 0.646 | 0.756 | 0.734 | 0.564 |
| SAM6D Lin et al. (2024) | | ✓ | 0.845 | 0.515 | 0.699 | 0.704 | 0.578 |

## 4.2 Evaluation of Object 2D Detection

**Seen Object 2D Detection.** YOLOX Ge et al. (2021) is a widely used advanced real-time object detection model that builds upon the YOLO Redmon et al. (2016) series. We follow the implementation of the SOTA method for object pose estimation GDRNet Wang et al. (2021) to train the YOLOX model on our synthetic training data and test on the real test split.

**Unseen Object 2D Detection.** CNOS Nguyen et al. (2023) is a model-based method that uses vision foundation models SAM Kirillov et al. (2023) and DINOv2 Oquab et al. (2023) for novel object segmentation and detection without re-training. It renders object templates from a CAD model and ranks SAM-generated segments by comparing their DINOv2 class token features with those of the templates. SAM-6D Lin et al. (2024) detects the objects with a similar strategy as CNOS Nguyen et al. (2023) but computes a weighted score including semantics, appearance, and geometry to match the query object template with the segments extracted from SAM Kirillov et al. (2023).

As shown in Table 4, for the 2D detection task, YOLOX Ge et al. (2021) is trained on the training split of each dataset and tests on the test split for the same objects (seen objects), achieving comparable results across different datasets. However, for the unseen object detection on our dataset, CNOS and SAM6D show a clear drop compared to other BOP datasets. These results highlight the increased difficulty of our dataset for 2D detection, due to heavy occlusion, repeated object instances, and strong surface reflections. We show a qualitative comparison in Figure 5.

## 4.3 Evaluation of Object 6D Detection

**Seen Object 6D Detection.** SurfEmb Haugaard & Buch (2022) learns per-object dense 2D–3D correspondence distributions over object surfaces using contrastive learning in an unsupervised fashion. It achieves strong performance on BOP and handles visual ambiguities effectively. GDRNet Wang et al. (2021) is a recent state-of-the-art framework that processes zoomed-in RoIs from RGB images to predict intermediate geometric features: dense 2D-3D correspondences, surface region attention maps, and visible object masks. These features guide a Patch-PnP module to directly regress the 6D pose in a differentiable manner.

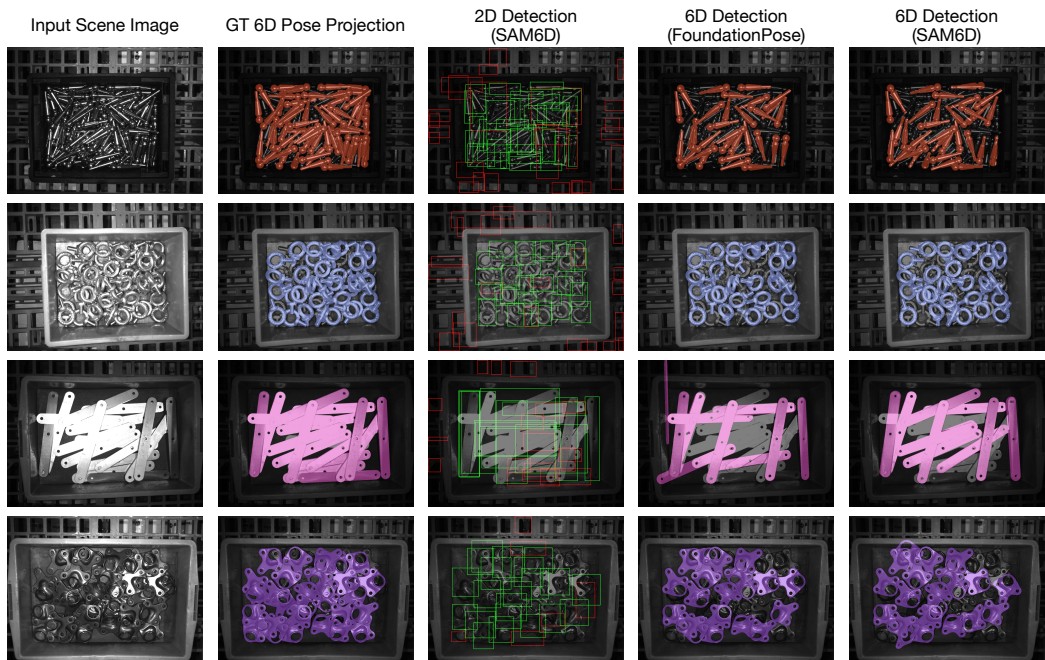

Figure 5: Example qualitative results of SOTA methods for unseen object 2D and 6D detection tasks on XYZ-IBD dataset. In the detection results, green boxes indicate correct object IDs, while red boxes indicate mismatches with the ground truth ID.

**Unseen Object 6D Detection.** FoundationPose Wen et al. (2024) supports both model-based and model-free settings using neural implicit representations for view synthesis. Trained on large-scale synthetic data with transformer-based coarse-to-fine design, it generalizes well and outperforms prior methods across benchmarks. SAM-6D Lin et al. (2024) uses the Segment Anything Model for segmentation and applies ViT Dosovitskiy et al. (2020) and GeoTransformer Qin et al. (2023) to extract features from RGB-D input and CAD models. Trained on a large-scale synthetic dataset, it achieves strong performance in model-based 6D pose estimation.

Unseen object methods assume the availability of object segmentation or detection as a prior for pose estimation. Accordingly, we use segmentation masks produced by SAM-6D for fair comparison among these methods, while seen object methods utilize detection results from YOLOX. As shown in Table 4, all methods struggle on our dataset. Specifically, both GDRNet and SurfEmb, representing seen object methods, fail to predict accurate poses, despite being trained on synthetic data. In contrast, unseen object methods demonstrate relatively better performance, with SAM6D achieving state-of-the-art results. Compared to existing household datasets, Van Nguyen et al. (2025); Sundermeyer et al. (2023), our dataset introduces greater challenges for pose estimation due to the complexity of object materials, geometric variations, and severe scene clutter. Figure 5 shows a qualitative comparison of the baseline results on XYZ-IBD benchmark.

## 5 CONCLUSION AND LIMITATIONS

we introduce the XYZ-IBD dataset, a high-precision bin-picking benchmark that captures real-world industrial-grade complexity, including object reflectivity, scene clutter, and heavy occlusion. The dataset comprises several industrial parts collected under real factory conditions using three different sensors, resulting in 273k real-world, annotated samples, along with a 45k-frame synthetic dataset simulating realistic bin-picking environments. Through a multi-stage, semi-automatic protocol, XYZ-IBD provides accurate 6D pose annotations, with error quantified via simulations that model real-world sensor and calibration noise, achieving pose errors as low as 1 mm. We believe XYZ-IBD brings real-world industrial vision problems to the academic community and helps bridge the gap between academic research and practical application. While we focus on a specific industrial scenario for bin-picking, the working distance and the scale of the objects are still limited, which is a potential limitation for the methods to generalize to other objects with different materials and shapes.

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

# Appendix

## A DATA COLLECTION HARDWARE SETUP

Figure 6 shows the robotic setup and camera configuration on the robot arm. We mount three cameras with different resolutions, working distances, and depth technologies. The cameras are mounted on the end of the robotic arm, collecting the scenes synchronously. We compare the sensor's configuration in Table 5.

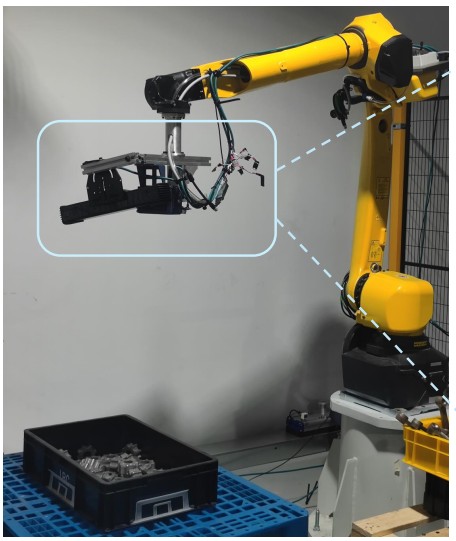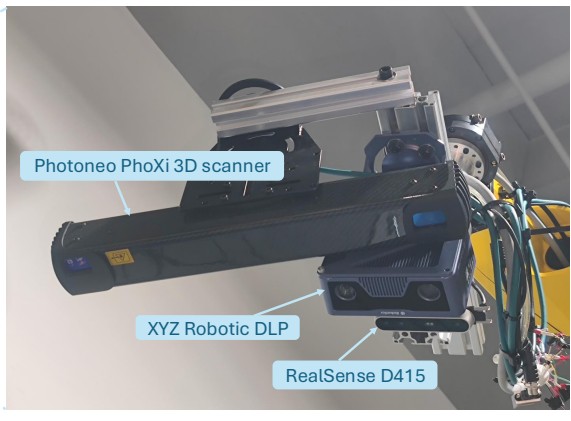

Figure 6: The data collection setup for the robot arm and sensors.

| Attribute | Working Distance (mm) | Resolution | Modalities | Depth Technology |
|---|---|---|---|---|
| **RealSense D415** | 500 − 3000 | 1920 × 1080 | RGB/Depth | Stereoscopic |
| **XYZ Robotics AL-M DLP** | 350 − 800 | 1440 × 1080 | Grayscale/Depth | Structured Light |
| **Photoneo PhoXi M** | 458 − 1118 | 2064 × 1544 | Grayscale/Depth | Laser Scanner |

Table 5: Configuration of the sensors.

## B SYNTHETIC TRAINING DATA

All collected industrial objects are used to generate the synthetic training dataset. We show the CAD models and the real objects for the collected industrial parts in Figure 8. For each scene, we simulate a free-fall of multiple object instances and render 25 images under varying lighting conditions and material properties. The rendering process uses the same camera intrinsics as the XYZ Robotics structured light camera. For each object, 120 scenes are rendered, resulting in approximately 3,000 bin-picking frames per object. This bin-picking synthetic dataset provides ground truth object masks, depth images and object 6D poses, therefore can be used as the training set for the depth estimation, 2D detection and pose estimation tasks. In total, the synthetic training dataset contains 45,000 RGB-D frames and occupies about 80 GB.

## C BENCHMARK SETUP

For each of the 15 real industrial parts, we collected 5 different scenes by varying the number of instances, lighting conditions and object poses. We use 1 scene for each object as the validation set, and 4 scenes for each object as the test set. In the test set, we follow the BOP Challenge 2025's setup and provide both single-view and multi-view evaluation protocols. For multi-view evaluation,

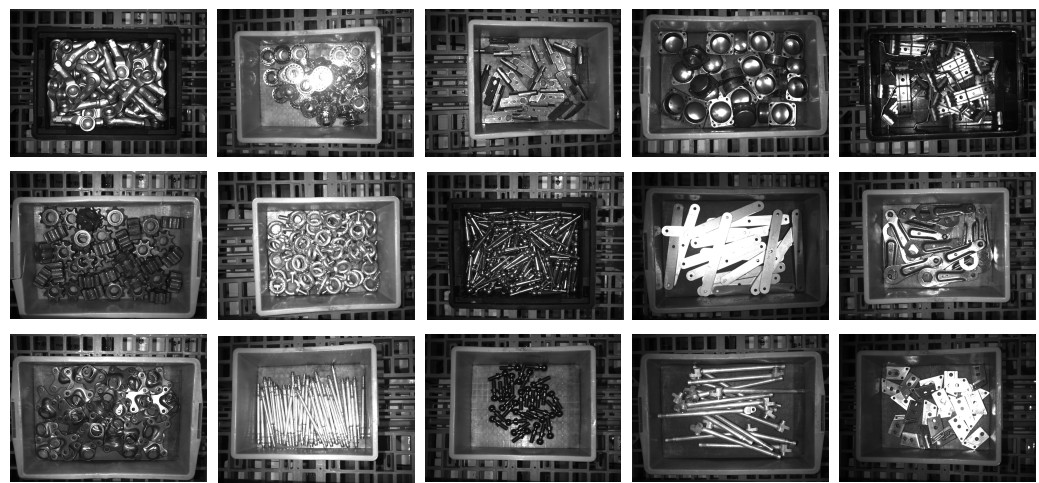

Figure 7: Example scenes in XYZ-IBD dataset.

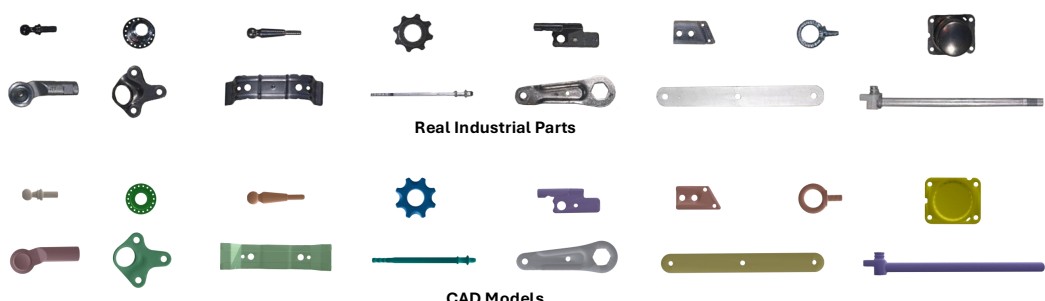

Figure 8: The collected industrial parts and their corresponding CAD models of XYZ-IBD dataset.

5 viewpoints are selected per scene based on the maximum spatial spread. One view is designated as the target view for evaluation, while the other 4 views serve as reference views. Relative camera poses among the 5 views are provided to enable multi-view methods to exploit spatial context. For single-view methods, only the target view is used. The ground truth pose of the validation set is released publicly, but the ground truth of the test set is hidden and hosted in the BOP evaluation system. The validation set size is approximately 8 GB, while the test set occupies around 3 GB.

## D    IMPLEMENTATION DETAILS FOR THE BASELINES

All seen-object baseline methods are trained on our synthetic training dataset and evaluated on the real testing split. For all the unseen baselines, we directly use the pretrained model to infer on the testing split.

### D.1    YOLOX

We train a YOLOX Ge et al. (2021) model for object detection following the configuration used in GDRNPP Liu et al. (2025). Training is performed on a single NVIDIA RTX 4090 GPU with a batch size of 24 for 30 epochs. Data augmentation is applied during the first 15 epochs, consistent with the GDRNPP setup. The complete training process on the synthetic PBR dataset takes approximately 18 hours.

### D.2 GDRNET

GDRNet Wang et al. (2021) is trained on all the objects of our synthetic dataset with a batch size of 24 for 10 epochs, totaling roughly 490,000 training steps. The training is conducted on a single NVIDIA RTX 4090 GPU and completes in approximately 24 hours.

### D.3 SURFEMB

SurfEmb Haugaard & Buch (2022) is trained on each object of our synthetic dataset with a batch size of 24 for 500,000 steps. Training is performed on a single NVIDIA RTX 4090 GPU and takes approximately 20 hours.

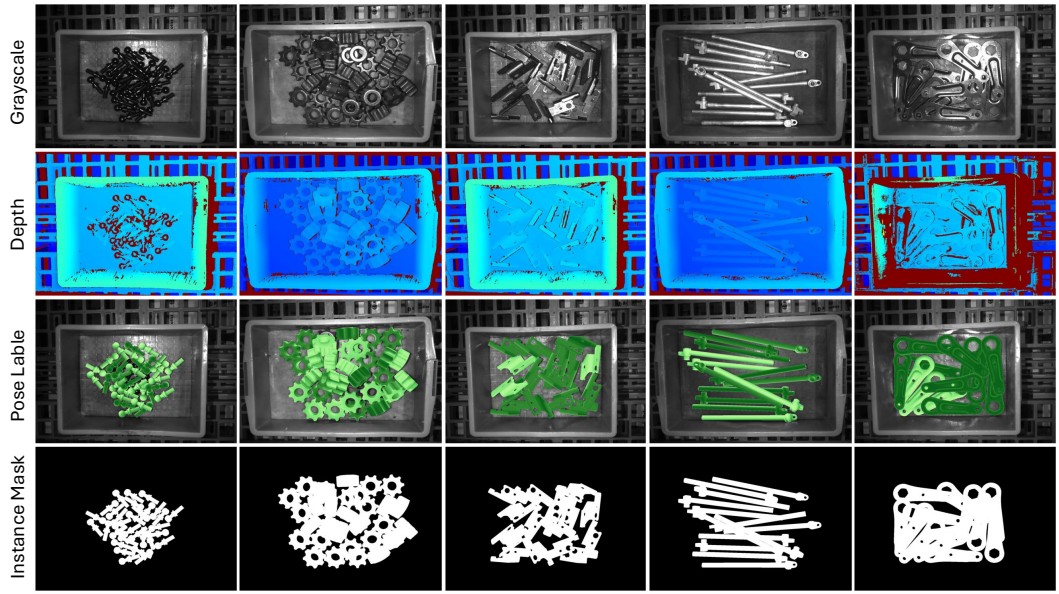

Figure 9: More data samples from the XYZ camera.

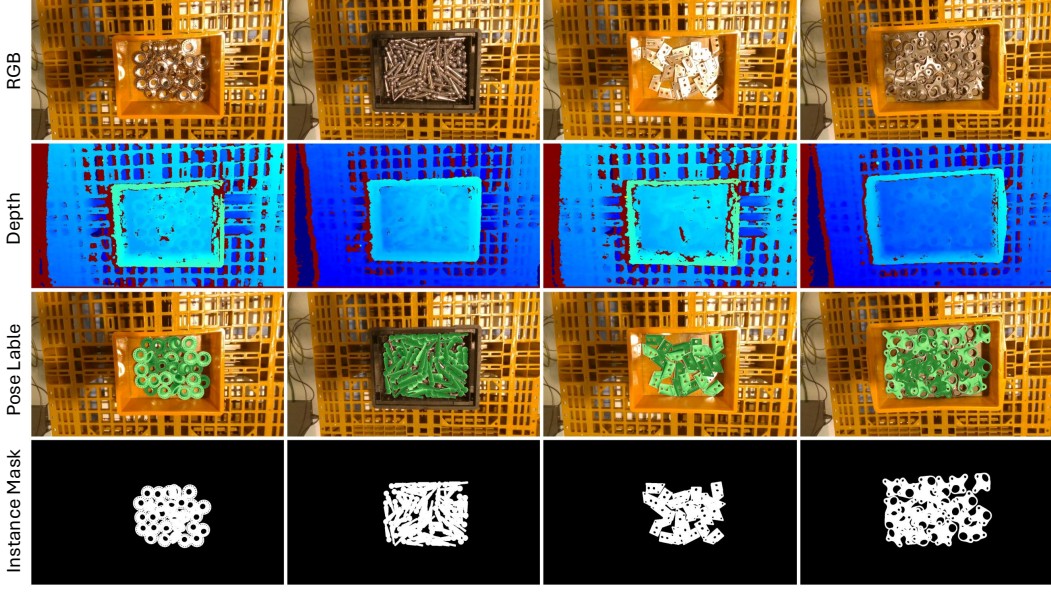

Figure 10: More data samples from the RealSense camera.

# E MORE VISUALIZATIONS OF DATA SAMPLES

We visualize more data samples in this section. Figure 9 shows more examples for the scenes that were recorded with the XYZ camera, and Figure 10 shows more examples from the RealSense camera. We compare our dataset with other BOP datasets with the instance distribution in Figure 11. The dataset follows the BOP dataset format as shown in Figure 12.

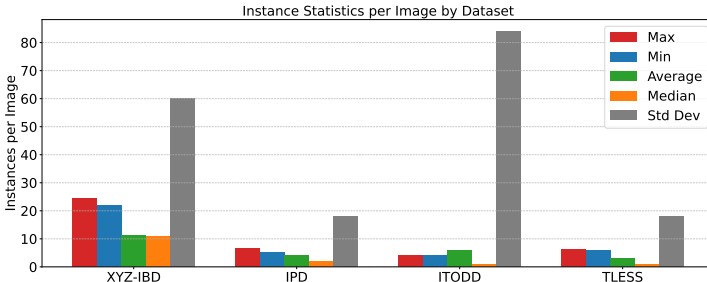

Figure 11: The instance distribution of the BOP industrial datasets.

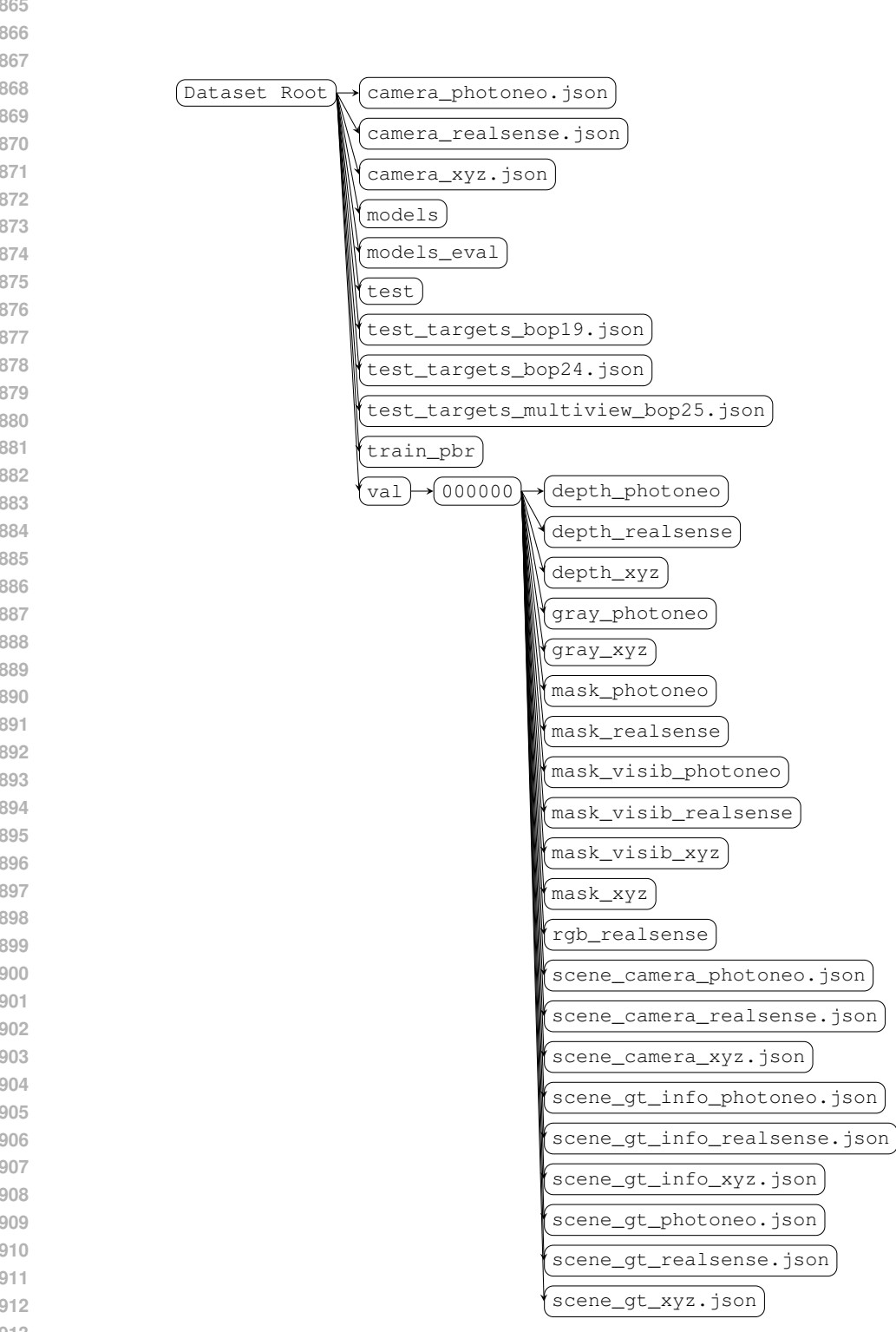

Figure 12: Directory structure of the dataset.