# OpenReview forum: "XYZ-IBD: Benchmarking Robust 6D Object Pose Estimation under Real-World Industrial Complexity"
_ICLR.cc/2026/Conference — ICLR 2026 Conference Withdrawn Submission_

### Official Review · Reviewer_9ZYz · 2025-10-30

**Soundness:** 3
**Presentation:** 1
**Contribution:** 2
**Rating:** 2
**Confidence:** 4

**Summary:**

The authors introduce a new dataset for benchmarking 6DOF pose estimation methods. The dataset focuses on industrial objects, with realistic contexts, such as objects are cluttered, densely stacked. Experiments are performed and compared with other benchmarks for a number of SOTA methods.

**Strengths:**

- I fully agree with the authors that more realistic, more challenging benchmarks are relevant for research in industrial applications.
- The dataset seems valuable, is well documented and should have a positive impact on the research field
- Recent SOTA methods are evaluated on the benchmark, and compared with existing benchmarks.

**Weaknesses:**

- The paper is poorly written and needs some extra attention, see below for more detailed feedback.
- As I said, I think this is very relevant research, but I doubt the fit with the ICLR venue. I think the scope of the dataset is too narrow and too applied.

**Questions:**

- Please use \citep (parentheses)
- The bright red and green for resp. citations and fig. referencing is hard to read. I suggest to use darker colors, or black.
- The XYZ-IBD acronym is not explained in the abstract. What does the XYZ stand for? It seems redundant.
- Abstract: "which approach saturation", unclear what this means exactly.
- line 37: "segment and estimate"
- Fig.2 is a bit unclear, provide more context in the caption
- line 214: "sizes ("
- inconsistent use of spaces between number and unit (e.g. line 232)

---

### Official Review · Reviewer_WsAo · 2025-10-31

**Soundness:** 2
**Presentation:** 2
**Contribution:** 3
**Rating:** 4
**Confidence:** 4

**Summary:**

This paper presents the XYZ-IBD dataset for 6D object pose estimation in industrial bin-picking scenarios. The work aims to capture challenging industrial conditions including reflective materials, severe occlusions, and dense clutter. The dataset construction shows effort, and the multi-sensor setup integrating RGB, grayscale, and depth modalities is reasonable. The work addresses a practical scenario that could be relevant for robotics applications.

**Strengths:**

The dataset construction demonstrates considerable effort and technical competence. The multi-sensor data acquisition setup integrating RGB, grayscale, and depth modalities is well-designed and provides diverse data modalities that could benefit future research in industrial pose estimation. The semi-automatic annotation pipeline with multiple validation steps shows careful attention to data quality. The high annotation accuracy of 0.99mm, while validated through simulation, represents a notable achievement for industrial applications requiring millimeter-level precision.
The work addresses a practical industrial scenario that appears underrepresented in existing benchmarks. Bin-picking with industrial objects presents real challenges including reflective materials, severe occlusions, and dense clutter, which are relevant for robotics applications. The dataset includes both real-world and synthetic data, providing training resources for the research community.
The benchmarking effort with multiple state-of-the-art methods provides baseline results that could serve as useful references. The inclusion of both seen and unseen object evaluation protocols demonstrates consideration of different use cases.

**Weaknesses:**

First, the paper does not clearly establish what makes XYZ-IBD uniquely valuable compared to existing industrial datasets such as T-LESS, ROBI, and IPD. The core challenges addressed have been partially covered by these prior works. The distinction from recent work like IPD 2024, which also employs multi-sensor approaches for similar scenarios, remains unclear. While XYZ-IBD achieves higher annotation accuracy, this appears to be an engineering improvement rather than a conceptual advance.
Second, the experimental validation appears limited. The benchmarking includes only a constrained set of mainstream methods and lacks comparisons with industrial-specific pose estimation frameworks. The performance degradation observed is not attributed to specific factors. The results would benefit from error analysis categorizing failures by object type, occlusion level, and material properties.
Third, key technical details are not sufficiently documented. The specific implementation of multi-view depth fusion and ICP refinement in the annotation pipeline is not clearly described. The synthetic dataset generation process, particularly how physics-based interactions are simulated, lacks detail needed for reproducibility. The validation method relying on simulated environments may have limitations due to domain gaps with reality.
Fourth, the use of anti-reflection spray during data collection raises concerns about whether the dataset truly represents unmodified industrial environments. This practice fundamentally alters optical properties and cannot be applied in real industrial testing scenarios, which undermines the practical utility of the benchmark.
Fourth, the discussion of limitations is brief. The paper mentions constraints on working distance and object scale but does not explore how these affect applicability to other industrial scenarios. Potential biases in data collection are not adequately analyzed.

**Questions:**

(1)	Compared to existing industrial datasets such as T-LESS, ROBI, and IPD, how does XYZ-IBD uniquely address unmet challenges through its object selection, scene design, and annotation accuracy?
(2)	Why were only a limited number of mainstream methods selected for benchmarking? How might results differ if compared with industrial-specific frameworks optimized for reflective and texture-less objects?
(3)	Could you provide detailed implementation details of multi-view depth fusion and ICP refinement to ensure reproducibility?
(4)	What measures were taken to mitigate biases during data collection? How might fixed lighting conditions and object arrangement affect model generalizability?

---

### Official Review · Reviewer_Z2mu · 2025-11-01

**Soundness:** 2
**Presentation:** 2
**Contribution:** 2
**Rating:** 4
**Confidence:** 3

**Summary:**

The paper presents XYZ-IBD, a new benchmark dataset for 6D pose estimation in industrial bin-picking scenarios. The dataset aims to capture the visual and geometric complexity of real-world industrial settings, which are not well represented in existing benchmarks.

**Strengths:**

The paper presents a contribution through the introduction of a new benchmark, XYZ-IBD, designed specifically for realistic industrial bin-picking scenarios. It addresses a domain that existing household-oriented datasets fail to capture. The authors combine RGB, grayscale, and depth modalities from multiple industrial-grade sensors while the data collection and annotation pipeline is of high quality which could be observed from the supplementary materials provided. The work is overall well-written and well-organized.

**Weaknesses:**

The paper should provide more baselines such as MegaPose which has recently emerged as a strong generalizable baseline trained on massive synthetic datasets with extensive domain randomization, designed precisely for cross-domain and industrial-level robustness. Given that XYZ-IBD explicitly aims to test generalization to real-world industrial conditions, the absence of such a model makes it difficult to assess whether the benchmark truly challenges state-of-the-art generalizable pipelines. Incorporating it would provide a more comprehensive performance landscape and help quantify whether industrial failures are due to dataset complexity or method generalization limits.

The current paper clearly motivates the need for industrial realism but does not explicitly articulate the scientific questions or hypotheses that XYZ-IBD aims to test. As a result, the benchmark’s intended research contribution is somewhat implicit rather than formally defined. One format is that "RQ1: Can synthetic data augmentation (e.g., BlenderProc) sufficiently bridge the domain gap for reflective, texture-less objects?"

**Questions:**

The paper should compare with DTTD[1] & DTTD2[2] which aims for industrial indoor scenarios using commercial and iPhone sensors.

[1] Feng, Weiyu, et al. "Digital twin tracking dataset (dttd): A new rgb+ depth 3d dataset for longer-range object tracking applications." Proceedings of the IEEE/CVF Conference on Computer Vision and Pattern Recognition. 2023.
[2] Huang, Zixun, et al. "Robust 6DoF Pose Estimation Against Depth Noise and a Comprehensive Evaluation on a Mobile Dataset." Proceedings of the Computer Vision and Pattern Recognition Conference. 2025.

---

### Official Review · Reviewer_DE1B · 2025-11-02

**Soundness:** 2
**Presentation:** 3
**Contribution:** 3
**Rating:** 4
**Confidence:** 3

**Summary:**

This paper introduces a new benchmark dataset named XYZ-IBD for 6D object pose estimation, specifically for industrial bin-picking applications. It provides both RGB and depth data. It uses a semi-automatic annotation pipeline. It also provides physics-based simulations to generate synthetic data. It also provides benchmarking of several SOTA 6D pose estimation methods to show that their performance will significantly drop compared to other 6D object pose estimation benchmarks.

**Strengths:**

- This paper addresses a real industrial problem of bin picking of a large number of objects in the same shape. The problem exhibit sufficient occlusion, reflective surface, and clutter which are challenging for robotic perception.
- The authors argued a <1mm annotation accuracy, which is impressive.
- The benchmarking of several SOTA methods shows bad results. It means that the dataset maybe have introduced an unsolved challenging problem that previous datasets were not able to address.

**Weaknesses:**

- The annotation process is in extremely high cost. The annotators need to manually align 3D CAD models to each object instance in the image via a GUI which is very slow and costly. This is probably the main reason that this dataset is limited in scale.
- It is unsure how the authors measures the average annotation error correctly, especially when the depth is generated by projected pattern deformation analysis. The depth generation process involves a lot of complex steps and it is hard to judge if the error measurement is correct.
- The authors applied anti-reflection spray to the objects before collecting the data. This is not OK in my opinion. Anti-reflection spray is indeed good for capturing depth on which the pose annotation relies. However, in real applications, there is no anti-reflection spray applied. The depth sensor signal will be very different, meaning that all the pose data collected with anti-reflection spray will be meaningless. On the other hand, without anti-reflection spray, there is no way to accurately annotate the pose data.
- The main data for high-precision annotation is collected by industrial-grade high-precision scanners. This limits the application scope of this dataset, since not everyone has the same scanner in their lab as the authors. Though Intel Realsense cameras are also used to collect data, however, I highly doubt if the Realsense is enough to provide sensing signal sufficiently accurate for pose estimation of these small objects. This is probably the reason why the SOTA methods show bad results.

**Questions:**

The authors can respond to the "Weakness" section of my review.

---

### Note · Authors · 2026-03-05

I have read and agree with the venue's withdrawal policy on behalf of myself and my co-authors.

---

### Meta-Review · Area_Chair_6YAt · 2026-01-06

**Summary:**

Reviewers collectively identified a crucial flaw in the dataset construction: the use of anti-reflection spray, which modifies the optical properties of the target objects, thereby nullifying the benchmark's claim of representing challenging, reflective industrial environments (DE1B, WsAo). Beyond this validity issue, reviewers criticized the lack of clear differentiation from existing benchmarks like T-LESS and IPD (WsAo, Z2mu), the omission of critical state-of-the-art baselines such as MegaPose (Z2mu), and the limited scalability of the manual annotation pipeline (DE1B), with Reviewer 9ZYz further questioning the paper's writing quality and fit.

**Reviewer Concerns:**

Since no author rebuttal was provided, all reviewer concerns remain outstanding.

**Reviewer Scores:**

In the absence of a rebuttal and given the critical nature of the flaws, reviewers would likely lower their scores toward a stronger rejection.

---

### Decision · Program_Chairs · 2026-01-26

Reject